# The Apple of Daddy’s Eye: Parental Overvaluation Links the Narcissistic Traits of Father and Child

**DOI:** 10.3390/ijerph17155515

**Published:** 2020-07-30

**Authors:** Gabrielle Coppola, Pasquale Musso, Carlo Buonanno, Cristina Semeraro, Barbara Iacobellis, Rosalinda Cassibba, Valentina Levantini, Gabriele Masi, Sander Thomaes, Pietro Muratori

**Affiliations:** 1Department of Education, Psychology, Communication, University of Bari Aldo Moro, 70122 Bari, Italy; pasquale.musso@uniba.it (P.M.); cristina.semeraro@uniba.it (C.S.); barbara.iacobellis@uniba.it (B.I.); rosalinda.cassibba@uniba.it (R.C.); 2Scuola di Psicoterapia Cognitiva and Associazione di Psicologia Cognitiva, 00185 Rome, Italy; buonanno@apc.it; 3IRCCS Fondazione Stella Maris, Scientific Institute of Child Neurology and Psychiatry, 56128 Pisa, Italy; valentina.levantini@unifi.it (V.L.); gmasi@fsm.unipi.it (G.M.); 4Department of Psychology, Utrecht University, 3512 JE Utrecht, the Netherlands; s.thomaes@uu.nl

**Keywords:** childhood narcissistic traits, parental overvaluation, parenting, father’s narcissism, parenting

## Abstract

This study contributes to the literature on the parental correlates of children’s narcissism. It addresses whether parental overvaluation may drive the putative link between parents’ narcissism and children’s narcissism and self-esteem. The cross-sectional design involved a community sample of 519 school-age children (age ranging from 9 to 11 years old) and their parents from an Italian urban context. Child-reported measures included narcissistic traits and self-esteem, while parent-reported measures included narcissistic traits and overvaluation, as well as parenting styles. A series of structural equation models, run separately for mothers and fathers, showed that both parents’ narcissism was directly and positively related to overvaluation and the children’s narcissistic traits; overvaluation partially mediated the indirect link between the fathers’ and children’s narcissistic traits. None of the parenting-style dimensions were related to the children’s outcomes, with the exception of the mothers’ positive parenting being directly and positively related to children’s self-esteem. These findings shed new light upon the parental correlates of child narcissism by suggesting that mothers and fathers convey their narcissism to their offspring through differential pathways. Our findings may be understood from universal as well as cultural specifics regarding the parenting roles of mothers and fathers. Clinical implications for the treatment of youth narcissism suggest the potential of targeting not only children but also their parents.

## 1. Introduction

Narcissism has long fascinated clinicians and researchers and still does today. While early work focused on narcissism as a personality disorder in adults, narcissism is currently mostly studied as a dimensional trait, in adults as well as in children. Different from what common belief suggests, narcissism is not the same as excessive self-esteem. In fact, narcissism and self-esteem typically are only weakly related [1,2,3,4], despite some inconsistencies [5]. Individuals with narcissistic traits view themselves as superior to others, but they are not necessarily satisfied with themselves. The presence of narcissistic traits puts children and adolescents at risk for the development of emotional and behavioral problems. More specifically, youth narcissism is associated with both aggression and conduct problems, as well as with anxiety and depressive symptoms [1,6,7,8]. 

The problematic outcomes associated with childhood narcissism require new, tailored intervention approaches. These approaches need to be informed by a theory- and evidence-based understanding of the developmental origins and socialization of youth narcissism. 

Theoretical accounts of the etiology of narcissism have emphasized the importance of parenting practices and parent–child relationships [9,10,11]. Early socialization experiences, especially with primary caregivers, are crucial for healthy self-concept development. When parents are relatively unable to provide functional socialization experiences during the child’s early years of life, this may place children on a pathway of narcissistic maladjustment. Some theorists have argued that narcissism may be cultivated, specifically, by indulgent and permissive parenting, and the excessive use of unconditional praise [11,12]. They have argued that this type of parenting may lead children to think that they are special and entitled to a distinct treatment and, at the same time, may make them dependent upon external validation. In another account, Kernberg [9] suggested that narcissism may be caused by parents who look upon their child as special and talented but appear cold or strict, and hold excessively high expectations. In this context, parents are thought to provide praise and affection only when the child fulfills their high aspirations. When the child fails to do so, they hold back praise and affection. Here, children may develop narcissistic self-views to protect themselves from feelings of rejection. Yet another account has suggested that children’s narcissism may be cultivated by the parents’ narcissistic use of the child [13]. According to this view, the child is a tool for the parents’ fulfillment of their own motives and desires, at the expense of the healthy development of the child’s independent sense of self. Here, the emergence of the child’s narcissistic traits reflects an ongoing request for parental approval and validation.

Only recently, research has begun to empirically test the theoretical accounts of the socialization of narcissism. Research findings are still quite divergent. Parenting dimensions that have been associated with the developmental emergence of narcissism in youths include (maternal) permissiveness and authoritativeness [3], warmth and positive parenting, as well as psychological control, lack of monitoring, and inconsistent discipline [14,15,16]. Longitudinal findings in a large sample of youths also showed that both the mother’s and father’s hostility and lack of monitoring predicted their offspring’s narcissism over a period of two years [17].

As it stands, results suggest that different socialization strategies may contribute to children’s narcissistic traits, although the heterogeneity of the results may be partial due to methodological differences among studies. For example, studies have used somewhat different conceptualizations and operationalizations of youth narcissism, assessed conceptually different parenting dimensions, and differed in sample composition. Some studies have also suffered some methodological limitations, including the reliance on a single informant (e.g., [3,14,15,16]) or lack of control for the covariance existing between narcissism and self-esteem (e.g., [14]). Moreover, while most research has focused on adolescents or young adults, narcissism typically begins to develop already at an earlier age, in middle to late childhood [18]. Finally, a gap in the literature regards the question of whether the mother’s and father’s parenting may differentially relate to their children’s narcissism. Most studies have aggregated parenting measures across both parents (e.g., [15,16]). 

One promising line of research regarding the socialization of narcissism focused on parental overvaluation—parents’ tendency to consider their child as more special and deserving than others. In line with Social Learning Theory [19], it assumes that children who are overvalued by their parents are prone to develop narcissistic features. This may be especially true when parents use inflated, indiscriminate praise that predisposes children to consider themselves superior and more entitled than their peers. Brummelman et al. [20] longitudinally investigated the association between parental overvaluation and children’s narcissistic traits. Results showed that, in a large community sample of children (age 7–11 at the baseline assessment), both maternal and paternal overvaluation predicted increased narcissistic traits in children over time. Interestingly, parental overvaluation was not related to the children’s levels of self-esteem, which instead was predicted by parental warmth. In another study, Brummelman and colleagues [21] found that parental overvaluation is unrelated to basic parenting dimensions, such as warmth and control, but positively associated with parental narcissistic traits. Other studies have similarly found a link between overly positive, inflated praise and narcissistic traits in children (ages 7–11 years) who also have high self-esteem [22]. Children tend to value themselves based on others’ feedback and what they believe others think about them. So, when parents excessively praise them, this predisposes them to think they are extraordinary. This might be especially relevant for children with higher levels of self-esteem, who already hold a positive self-concept.

It is plausible, as hypothesized by Brummelman and colleagues [20], that parental overvaluation predicts children’s narcissistic traits because the child mimics or inherits the parents’ narcissistic traits. This suggestion is consistent with predictions from the developmental psychopathology framework, according to which parental psychopathology is a well-recognized risk factor for children’s mental health. Besides, genetic transmission and environmental factors, in particular those associated with the quality of parenting, may mediate the impact of parental psychopathology on child development [23]. Although this suggestion is theoretically well-founded and has been well-supported for various forms of parental psychopathology, it is still unexplored for parents’ (and consequently, their children’s) narcissistic traits.

Because clinical theory and empirical work have demonstrated the importance of the parent–child relationship during late childhood, the present study aims to further explore the parental correlates of children’s narcissistic traits. It does so by addressing several unanswered issues. Firstly, the present study tested the hypothesis that parental overvaluation would mediate the putative link between parental narcissistic traits and children’s narcissistic traits (but not children’s self-esteem). We also hypothesized that negative parenting would be associated with children’s narcissistic traits (but not self-esteem), while positive parenting would be associated with children’s self-esteem (but not narcissistic traits). We tested the models separately for mothers and fathers, although we did not formulate specific hypotheses, given inconsistent previous findings. We used children’s sex as a control variable in all models.

## 2. Materials and Method

### 2.1. Participants

Participants were fourth- and fifth-grade students from primary schools located in an urban region in south-eastern Italy (Apulia). The schools were located in five cities, reflecting diverse urban characteristics, and can be easily found in Italy (e.g., a regional capital, provincial capitals, peripheral areas of such capitals, and provincial towns). We included all participants for whom complete information could be obtained from at least one parent. The final child sample consisted of 519 participants (males = 47.6% and females = 52.4%) aged 9–11 years (*M* = 9.67, *SD* = 0.64). The mother sample consisted of 494 participants aged 27–55 years (*M* = 41.69, *SD* = 4.89), and the father sample consisted of 442 participants aged 31–70 years (*M* = 45.11, *SD* = 5.45). For most children (82.5%), the dataset contained information from both parents. The samples were homogeneous in terms of racial and ethnic composition, with at least 96% of children and parents being Italian Caucasian. About 95% of children had married or cohabiting parents, 4% separated or divorced parents, and 1% a single or widowed parent. About 16% of the children were also an only child, whereas 36% were first born, 38% second born, 9% third born, and 1% fourth born or later. The children’s parents were occupationally diverse (mothers: 9% skilled and technical workers, 40% tradespeople and service workers, 10% laborer and artisan, and 45% homemakers, unemployed, or other types of unskilled workers; fathers: 22% skilled and technical workers, 43% tradespeople and service workers, 31% laborers and artisans, and 4% homemakers, unemployed or other types of unskilled workers); 73% of mothers and 67% of fathers had completed their secondary schooling. These figures adequately represent the general Italian population of parents of fourth- and fifth-grade students in terms of occupation and education (see [24,25]).

### 2.2. Materials

All participants (children, mothers, and fathers) completed a paper-and-pencil survey containing a different set of measures. For the purposes of this study, a selection of these measures was used.

#### 2.2.1. Socio-Demographics 

Children were asked to indicate their sex (0 = male, 1 = female), age, and birth order. Parents were asked to indicate their sex, age, ethnicity, marital status, level of education, and occupation.

#### 2.2.2. Parenting 

The Alabama Parenting Questionnaire (APQ; [26], as adapted in Italy from Esposito, Servera, Garcia-Banda, and Del Giudice [27], was used to assess the mothers’ and fathers’ parenting style. The original version of the APQ, developed using USA samples includes 35 items measuring five domains of parenting: parental involvement and positive parenting (positive parenting scales), as well as poor monitoring/supervision, inconsistent discipline, and corporal punishment (negative parenting scales). Seven additional items evaluating the specific discipline practices other than corporal punishment are also usually included. However, this factorial structure was not replicated in the Italian context, where a simpler solution was found [27]. Specifically, the Italian version of the APQ consists of two scales based on a two-factor solution, namely positive parenting (12 items; e.g., “You hug or kiss your child when he/she does something well”) and negative parenting (7 items; e.g., “You feel that getting your child to obey you is more trouble than it’s worth”). Items are scored on a Likert-type scale ranging from 1 (never) to 5 (always). High scores are interpreted as adequate parenting practices for the positive scale and as inadequate parenting practices for the negative scale. The Italian validation study of the APQ showed satisfactory internal consistency reliabilities of the positive and negative scales for both mothers and fathers (all Cronbach’s alpha coefficients > 0.74). In this study, we excluded three of the seven items on the negative scale (e.g., “Your child stays out in the evening past the time he/she is supposed to be home”) because they are not suited for primary school children. Thus, we performed confirmatory factor analyses (CFAs) to test the factorial validity of our APQ measure (see the “Data Analysis” section for model fit criteria) using a parceling approach for latent modeling (see [28]) of the positive scale (this choice was aimed to limit the total number of observed indicators in the subsequent structural equation models estimated to meet our research aims). Specifically, separately for mothers and fathers, items for the positive parenting scale were parceled into four indicators comprising three items, with an equal distribution of factor loadings across parcels. Each parcel was computed by averaging the responses across the selected items. The CFAs, based on a robust maximum likelihood estimation procedure, supported the two-factor structure previously found by Esposito et al. [27], both for mothers (χ^2^(19) = 50.45, *p*< 0.001, CFI = 0.924, RMSEA = 0.058, SRMR = 0.061) and fathers (χ^2^(19) = 54.21, *p* < 0.001, CFI = 0.932, RMSEA = 0.065, SRMR = 0.068). The internal consistency reliability scores, calculated by the factor determinacy [29], were good for the positive parenting scale (0.86 for mothers and 0.90 for fathers) and adequate for the negative parenting scale (0.74 for mothers and 0.71 for fathers). Hence, this APQ measurement model was also used in our analyses.

#### 2.2.3. Parental Narcissism 

The 16-item version of the Narcissistic Personality Inventory (NPI-16; [30]), in its Italian adaptation (see [31]), was used to assess parental trait narcissism. The NPI-16 is a brief, unidimensional measure derived from the longer NPI-40 [32], designed to measure narcissism in nonclinical populations. It is composed of 16 pairs of forced-choice items, with each pair proposing two conflicting statements (one narcissistic, the other non-narcissistic). Participants express a preference for either the narcissistic (coded as 1, e.g., “I find it easy to manipulate people”) or the non-narcissistic response (coded as 0, e.g., “I don’t like it when I find myself manipulating people”). Higher scores are indicative of higher levels of narcissism. Adequate levels of face, internal, discriminant, and predictive validity were reported for the NPI-16 (e.g., [30,33]. To test its factorial validity in our mothers’ and fathers’ samples, we carried out CFAs using a parceling approach for latent modeling according to the procedure previously mentioned for the positive parenting scale of the APQ. Separately for mothers and fathers, items were parceled into four indicators comprising four items, with an equal distribution of factor loadings across parcels. Each parcel was computed by averaging the responses across the selected items. The CFAs, based on a robust maximum likelihood estimation procedure, supported the one-factor structure, both for mothers (χ^2^(2) = 0.80, *p* = 0.67, CFI = 1.00, RMSEA = 0.000, SRMR = 0.009) and fathers (χ^2^(2) = 2.55, *p* = 0.28, CFI = 0.998, RMSEA = 0.025, SRMR = 0.013). The factor determinacy scores were good (0.82 for mothers and 0.86 for fathers). Hence, this NPI-16 measurement model was also used in our analyses.

#### 2.2.4. Parental Overvaluation

The Parental Overvaluation Scale (POS) was used to assess “parents’ belief that their own child is more special and more entitled than other children” [21] (p. 666). It is a unidimensional measure comprising 7 items (e.g., “My child is more special than other children”) scored on a Likert-type scale ranging from 0 (not at all true) to 3 (completely true). Higher scores are indicative of the parents’ higher overvaluation of their children. The POS demonstrated high test–retest stability over 6, 12, and 18 months, as well as good convergent, discriminant, and criterion validity [21]. Since it was developed among Dutch and American parents, we needed to verify that it could be applied to Italian parents. We conducted exploratory (principal component) factor analyses using the oblimin rotation, for both mothers and fathers. Results suggested a one-factor solution explaining 38.63% and 41.94% of the total variance, respectively, for mothers and fathers. We further tested the factorial validity of the POS by CFAs using a parceling approach for latent modeling. Separately for mothers and fathers, six of the seven items were parceled into three indicators comprising two items, with an equal distribution of factor loadings across parcels. Each parcel was computed by averaging the responses across the selected items. The CFAs, based on a robust maximum likelihood estimation procedure, supported the one-factor structure, both for mothers (χ^2^(2) = 1.79, *p* = 0.41, CFI = 1.00, RMSEA = 0.000, SRMR = 0.011) and fathers (χ^2^(2) = 0.54, *p* = 0.76, CFI = 0.998, RMSEA = 0.025, SRMR = 0.006). The factor determinacy scores were good (0.87 for mothers and 0.88 for fathers). Hence, this POS measurement model was also used in our analyses

#### 2.2.5. Children’s Narcissism 

The Childhood Narcissism Scale (CNS; [6]), as validated by Muratori et al. [1] in the Italian context, was used to assess the children’s narcissism traits. It is a unidimensional measure including 10 items (e.g., “I think it’s important to stand out”) scored on a Likert-type scale ranging from 0 (not at all true) to 3 (completely true). Higher scores are indicative of higher levels of narcissism. The CNS demonstrated good internal consistency reliability as well as good convergent and discriminant validity [1,6]. To test its factorial validity in our sample of children, we conducted a CFA using a parceling approach for latent modeling. Items were parceled into four indicators comprising two or three items, with an equal distribution of factor loadings across parcels. Each parcel was computed by averaging the responses across the selected items. The CFA, based on a robust maximum likelihood estimation procedure, supported the one-factor structure (χ^2^(2) = 0.92, *p* = 0.63, CFI = 1.00, RMSEA = 0.000, SRMR = 0.008). The factor determinacy score was good (0.81). Hence, this CNS measurement model was also used in our analyses.

#### 2.2.6. Children’s Self-Esteem 

The Rosenberg Self-Esteem Scale (RSES [34]; Italian adaptation by Prezza, Trombaccia, and Armento [35] was used to assess children’s self-esteem. It is the most widely used one-factor measure to assess self-esteem, including 10 items (e.g., “On the whole I am satisfied with myself”; five items are reverse worded) scored on a Likert-type scale ranging from 0 (strongly disagree) to 3 (strongly agree). Higher scores are indicative of higher levels of self-esteem. Reliability and validity for the RSES have been provided in several studies (for recent meta-analytic studies, see [36,37]). To test its factorial validity in our sample of children, we conducted a CFA using a parceling approach for latent modeling. Items were parceled into four indicators comprising two or three items, with an equal distribution of factor loadings across parcels. Each parcel was computed by averaging the responses across the selected items. The CFA, based on a maximum likelihood estimation procedure, supported the one-factor structure (χ^2^(2) = 4.22, *p* = 0.12, CFI = 0.993, RMSEA = 0.046, SRMR = 0.016). The factor determinacy score was good (0.83). Hence, this RSES measurement model was also used in the following analyses.

### 2.3. Procedure

We explained the study’s purpose to the participating school principals and teachers, who provided their informed consent for the research. Parents were informed about the research purposes through a letter and we obtained their written informed consent prior to data collection. If parents consented to their child’s and their own participation, they were asked to complete four questionnaires (i.e., one for sociodemographic data and one for self-reported parenting, narcissistic traits, and overvaluation). Two questionnaires were provided for each participating child, as both the mother and the father were asked to complete the questionnaires, based on their own free choice. If they did not consent, they were asked to return non-completed questionnaires to school. Before completing the questionnaires, parents were asked to specify whether they were fathers or mothers, and to identify themselves by using an ID code combining the first three letters of their child’s name and surname.

We completed the data collection in children in a single session at school. Research assistants arranged a visit to classes, providing students whose parents agreed to participate with a package containing the survey (assessing self-esteem and narcissistic traits). We provided general information about the aims of the study and explained the confidentiality of the data. To protect confidentiality, participants were instructed to choose an identifying code by combining the first three letters of their names and surnames, and then to fill in the self-reports at their leisure, taking all the time they needed. Teachers were not present in the classroom, to make sure that the children did not experience their research participation as a school task. Children were also assured that their participation was voluntary and that they could decline to participate at any time. All children whose parents agreed to participate completed their survey.

Families received no compensation and were treated in accordance with the ethical standards outlined by the American Psychological Association and the Italian Association of Academic Psychologists (AIP, www.aipass.org). The protocol for the data collection and treatment was approved by the department’s ethics committee (ethics reference code: ET-20-06).

### 2.4. Data Analysis 

The data analysis proceeded in several steps. First, to verify the univariate and multivariate normality of the distributions and be able to choose the correct estimation method in the subsequent analyses, we computed descriptive statistics (means, standard deviations, and normality indices) for the main study variables in Statistical Package for the Social Sciences (SPSS) version 24 (IBM Corp., Armonk, NY, USA). 

Secondly, we computed bivariate correlations, separately for the mother–child and father–child dyads, using a structural equation model in *Mplus 7.2* (Muthén & Muthén, Los Angeles, CA, USA) [29]. This model included a control (sex), the specified latent variables (positive parenting, negative parenting, parental narcissism, parental overvaluation, children’s narcissism, and children’s self-esteem), and allowed covariances between them to be freely estimated.

Thirdly, to test the potential mediating role of parental overvaluation, we estimated a structural equation “full mediation” model. We fixed to zero the direct links between the independent variables (i.e., parental narcissism, as well as positive parenting and negative parenting) and the dependent variables (i.e., children’s narcissism and self-esteem) (see Figure 1).

Fourthly, because our hypotheses also suggested potential direct links between the independent and dependent variables, we additionally estimated six “partial mediation” models (see Figure 2 and the paths named a, b, c, d, e, and f). These models served to assess whether our proposed parental overvaluation mediator fully accounted for the putative associations between the independent and dependent variables.

In both the full mediation and partial mediation models, all indirect paths through parental overvaluation were tested, and each model was controlled for sex. Following Faraci and Musso [38] as well as Kline [39], multiple indices were used to evaluate the model fit (adopted cutoffs in parentheses): the chi-square (χ^2^) test value with the associated *p* value (*p* > 0.05), comparative fit index (CFI ≥ 0.90 for acceptable and ≥ 0.95 for good fit), root-mean-squared error of approximation (RMSEA ≤ 0.08 for acceptable and ≤ 0.05 for good fit), and standardized root mean square residual (SRMR < 0.10 for acceptable and ≤ 0.05 for good fit). However, by acknowledging the potential limitation of the χ^2^ test because of its sensitivity to reject the null hypothesis with large sample sizes and complex models [40], we mostly relied on the goodness-of-fit indices. Nested model comparison (the more restrictive vs. the less restrictive models) was used to examine whether direct paths of parental narcissism, positive parenting, and negative parenting with children’s narcissism and self-esteem were tenable. In this case, the criteria for ascertaining significant differences included significant χ^2^ differences (Δχ^2^ with *p* < 0.05).

## 3. Results

### 3.1. Preliminary Analyses

Table 1 summarizes the descriptive statistics. It shows how some variables were not normally distributed with skewness and kurtosis values >|1.00| [39]. As multivariate non-normality was also found (normalized Mardia’s coefficient was 13.24, *p* < 0.001 for the mother-child sample, and 9.79, *p* < 0.001 for the father-child sample), the data were subsequently analyzed using the maximum likelihood robust estimation method (MLR in context of *Mplus*). Table 2 reports the bivariate correlations among the latent and control variables after estimating the structural equation models specifying all covariances between them. Both the models including mothers and children (χ^2^(255) = 338.57, *p* = 0.0004, CFI = 0.952, RMSEA = 0.026, SRMR = 0.042) and fathers and children (χ^2^(255) = 361.52, *p* < 0.001, CFI = 0.943, RMSEA = 0.031, SRMR = 0.047) fit the data well.

### 3.2. Mediation Models

#### 3.2.1. Mother-Child Models

We estimated the full and partial mediation models. In the initial estimate of the full mediation model, we controlled for sex by allowing it to predict all the latent variables; however, sex was only significantly linked to negative parenting and children’s narcissism. Only these significant paths were retained, and the model was re-estimated. This full mediation model had good fit (χ^2^(266) = 354.50, *p* = 0.0002, CFI = 0.949, RMSEA = 0.026, SRMR = 0.046). Next, we estimated the six partial mediation models. We found no significant differences when comparing the full mediation and the partial mediation models, including the direct links from positive parenting (Δχ^2^(1) = 0.75, *p* = 0.32) or negative parenting (Δχ^2^(1) = 0.56, *p* = 0.45), to children’s narcissism, nor from negative parenting (Δχ^2^(1) = 0.56, *p* = 0.45) or mothers’ narcissism (Δχ^2^(1) = 0.15, *p* = 0.70) to children’s self-esteem. The comparison of the full and the partial mediation models, including the direct links from the mothers’ narcissism to children’s narcissism (Δχ^2^(1) = 5.77, *p* = 0.016) and from positive parenting to children’s self-esteem (Δχ^2^(1) = 4.44, *p* = 0.035), showed that the full mediation model fit the data significantly worse. Thus, the final (partial) mediation model included these paths (see Figure 3) (χ^2^(264) = 342.80, *p* = 0.0008, CFI = 0.954, RMSEA = 0.025, SRMR = 0.043). This model showed how the mothers’ narcissism was directly and positively related to the mothers’ overvaluation and children’s narcissism. It also showed how positive parenting was directly and positively related to children’s self-esteem. However, the mediator (mothers’ overvaluation) was not significantly associated with children’s narcissism and self-esteem. Moreover, no significant indirect associations were found through it. Finally, female children were more likely to have lower levels of narcissism and to experience negative parenting from mothers than male ones.

#### 3.2.2. Father-Child Models

We followed the same analytic steps as for the mother–child models. Only the significant path between sex and children’s narcissism was retained. The full mediation model had a good fit (χ^2^(267) = 370.189, *p* < 0.001, CFI = 0.945, RMSEA = 0.030, SRMR = 0.049). We found no significant differences when comparing the full mediation and the six partial mediation models (in all cases the value of Δχ^2^ with one degree of freedom was <2.41 and *p* > 0.12). Thus, we considered the full mediation model the final one (see Figure 4). This model showed how the fathers’ narcissism was directly and positively related to the fathers’ overvaluation, which, in turn, was directly and positively associated with the children’s narcissism. Furthermore, the indirect effect of the fathers’ narcissism on children’s narcissism through the mediation of the fathers’ overvaluation was significant and positive (β = 0.06, *p* = 0.03). Finally, female children were more likely to have lower levels of narcissism than males.

## 4. Discussion

This study aimed to contribute to our understanding of the parental correlates of children’s narcissistic traits by addressing several gaps in the literature. We sampled children in the developmental stage of late childhood, an age period that is still underrepresented in this research field despite its importance for the development of self-views. We assessed both parental overvaluation and parenting styles to get a comprehensive view of the socialization experiences associated with children’s narcissistic traits. Perhaps most importantly, and in line with the developmental psychopathology framework, we sought to contribute understanding of the putative link between parents’ narcissistic traits and those of their children. In doing so, we explored the differential effects for mothers and fathers.

We found that the positive link between fathers’ narcissistic traits and those of their children was partially mediated by the fathers’ overvaluation. This was different for mothers. The mothers’ narcissistic traits were directly and positively related to those of their children, but this link was not mediated by overvaluation. While both parents’ narcissism is associated with their children’s narcissism, this link might be driven by different developmental pathways. Speculatively, this may be because of the different roles that a mother and father tend to play in raising their children. Although such different roles are likely to differ across cultures and time, and parents’ involvement with their children is obviously not predetermined by their sex, different involvement of fathers and mothers in raising their children can be consequential. Fathers have been thought to play a particularly important role in introducing their children to the social world and helping them to navigate it [41]. This might leave room for fathers to play an important role in influencing the way children interact with and perceive themselves in relation to other people—potentially cultivating an interpersonal style marked by entitlement and superiority in children. Indeed, fathers’ narcissistic traits promote an excessively positive view of their children, which may manifest in terms of them modelling their children to interact with others with a sense of grandiosity. 

Mothers more typically play a role as primary source of well-being, security, and attunement to children’s needs. At the risk of maintaining cultural stereotypes, this may be particularly true for South Italian mothers. Previous findings obtained in the same cultural group show that mothers are more invested than fathers in child caregiving and take a large share of the responsibility for intergenerational continuity [42]. It is possible that the intergenerational continuity of narcissism from mothers to children is primarily due to the quality of affective exchanges and caregiving behaviors that were not the focus of the present study. We encourage further research to test these preliminary accounts of how Italian fathers and mothers may socialize narcissism in their children. 

The correlates of children’s self-esteem were largely different from those of children’s narcissism. Specifically, while children’s self-esteem was not associated with parental overvaluation, it was associated with a mother’s positive parenting, as expected. This is consistent with previous work suggesting that self-esteem is cultivated by parental warmth and acceptance [20,21], while the excessive use of inflated praise can have the opposite effect [22]. Indeed, according to the self-deflation hypothesis, inflated praise may implicitly convey to children that they should continue to meet very high standards and expectations. As it is impossible to reach these standards all the time, inflated praise may lead children to become dissatisfied with themselves when routinely used by parents.

Interestingly, our results also showed that boys tended to be more narcissistic than girls. This is consistent with findings reported on both children [6,20] and adults [43], and with previous work suggesting that male children view themselves more favorably and are more socially dominant (e.g., [44,45]).

Importantly, the study findings need to be interpreted in the light of a number of limitations. Firstly, our study design does not allow for teasing apart the genetic and environmental influences on the overlap between the parents’ and children’s narcissistic traits. We are also unable to draw conclusions about the direction of the effects. It is possible, for example, that children’s narcissistic traits may influence their parents’ traits and parenting behaviors. Longitudinal research is needed to test potential bidirectionality. Secondly, our findings derive from a healthy community sample rather than a clinical sample; generalizability to clinical populations cannot be assumed. Thirdly, we assessed parenting with self-report measures administered to parents. Such measures can be influenced by social desirability and dissimulation, perhaps especially so among parents with higher narcissistic traits. Further studies would benefit from the use of complementary parenting measures, including observational and child-reported measures. 

## 5. Clinical Implications

Based on our findings, we see most value in intervention approaches that target parents as well as their children: both can be supported so that children will increasingly derive feelings of worth and accomplishment from effortful behaviors, rather than from “fixed” abilities or performances. This should keep children from developing vulnerable self-images if they fail to live up to high expectations [22,46]. Acceptance and commitment therapy may provide a useful intervention framework to help children and their parents develop psychological flexibility and adaptive behavior through committed action in accordance with their personal values—an approach that seems promising for the treatment of narcissism [47,48,49].

## 6. Conclusions

Notwithstanding some limitations, our findings provide a new understanding of the parental correlates of children’s narcissistic traits. Parental narcissism may facilitate the development of narcissistic traits in children. In fathers, this is partly due to narcissistic fathers being more likely to overvalue their children. Our findings inform the development of tailored intervention for youth narcissism and the availability of brief measures to assess parental overvaluation may help identify those fathers that could benefit from parenting support. 

## Figures and Tables

**Figure 1 ijerph-17-05515-f001:**
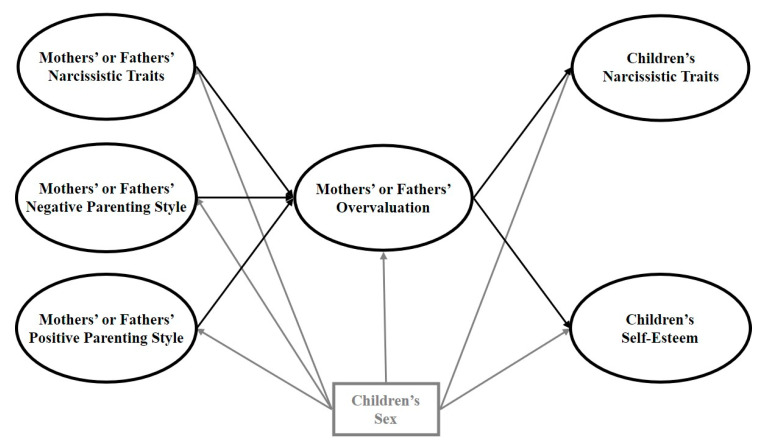
Theoretical full mediation model of the relations of parental narcissism, positive parenting, and negative parenting with children’s narcissism and self-esteem via parental overvaluation. The key study variables and their related paths are presented in black. As the control variable, sex and its related paths are presented in gray.

**Figure 2 ijerph-17-05515-f002:**
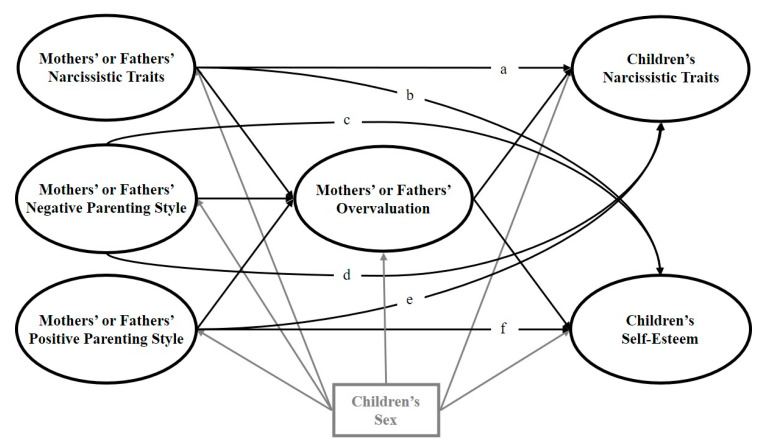
Theoretical partial mediation models of the relations of parental narcissism, positive parenting, and negative parenting with children’s narcissism and self-esteem including (1) the six direct paths a, b, c, d, e, or f; and (2) the indirect paths via parental overvaluation. The key study variables and their related paths are presented in black. As the control variable, sex and its related paths are presented in gray.

**Figure 3 ijerph-17-05515-f003:**
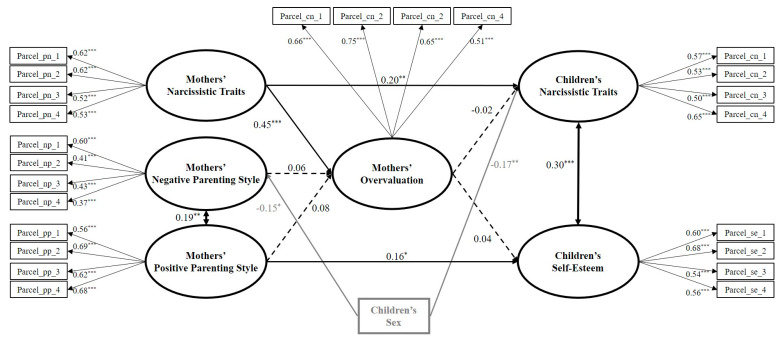
Estimated structural equation model for the best fitting mediation model among the mother–child dyads. *N* = 494. Standardized regression coefficients (betas) are shown. The measurement part of the model (including the parcels and factor loadings) is presented in black. The key study latent variables and their related paths are presented in bold black. Sex, as the control variable, and its related paths are presented in bold gray. Solid lines represent significant paths and dashed lines represent non-significant paths. For better visualization, residuals and non-significant covariances are not shown. * *p* < 0.05, ** *p* < 0.01, *** *p* < 0.001.

**Figure 4 ijerph-17-05515-f004:**
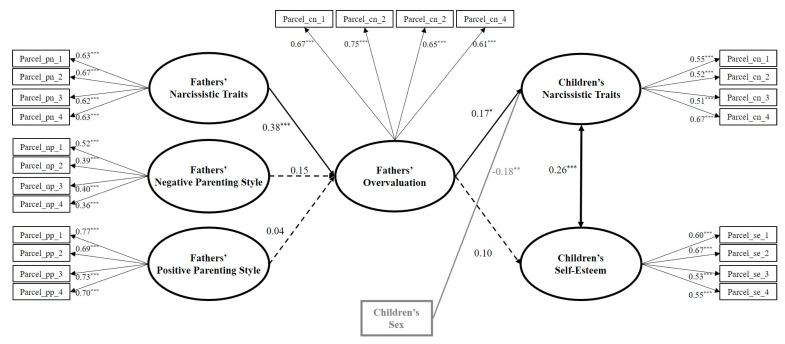
Estimated structural equation model for the best fitting mediation model among the father–child dyads. *N* = 442. Standardized regression coefficients (betas) are shown. The measurement part of the model (including the parcels and factor loadings) is presented in black. The key study latent variables and their related paths are presented in bold black. Sex, as the control variable, and its related paths are presented in bold gray. Solid lines represent significant paths and dashed lines represent non-significant paths. For better visualization, residuals and non-significant covariances are not shown. The standardized indirect effect of the fathers’ narcissism on children’s narcissism through the fathers’ overvaluation was β = 0.06, *p* = 0.03. * *p* < 0.05, ** *p* < 0.01, *** *p* < 0.001.

**Table 1 ijerph-17-05515-t001:** Mean scores, standard deviations, skewness, and kurtosis of the observed variables for the mother–child and father–child groups.

Observed Variable	Mean	Standard Deviation	Skewness	Kurtosis
Mother–child group (N = 494)				
Positive parenting (pp; scored 1–5)				
Parcel_pp_1	4.33	0.54	−0.73	0.16
Parcel_pp_2	3.76	0.59	−0.34	0.29
Parcel_pp_3	4.33	0.60	−1.05	0.86
Parcel_pp_4	4.20	0.62	−1.09	2.41
Negative parenting (np; scored 1–5)				
Parcel_np_1	2.65	1.21	0.20	−0.82
Parcel_np_2	2.16	1.19	0.70	−0.50
Parcel_np_3	2.60	1.12	0.09	−0.63
Parcel_np_4	1.74	0.93	0.82	−0.59
Parental narcissism (pn; scored 0–16)				
Parcel_pn_1	0.72	0.79	0.85	0.14
Parcel_pn_2	0.97	0.88	0.56	−0.42
Parcel_pn_3	0.50	0.77	1.66	2.71
Parcel_pn_4	0.49	0.73	1.45	1.61
Parental overvaluation (po; scored 0–3)				
Parcel_po_1	0.72	0.60	0.75	0.40
Parcel_po_2	0.75	0.64	0.98	0.74
Parcel_po_3	1.24	0.81	0.40	−0.53
Parcel_po_4	0.56	0.84	1.48	1.43
Children’s narcissism (cn; scored 0–3)				
Parcel_cn_1	0.67	0.55	1.00	1.22
Parcel_cn_2	0.83	0.72	0.77	0.14
Parcel_cn_3	1.23	0.76	0.36	−0.55
Parcel_cn_4	1.01	0.53	0.54	0.37
Self-esteem (se; scored 0–3)				
Parcel_se_1	1.82	0.70	−0.31	−0.31
Parcel_se_2	2.05	0.59	−0.33	−0.21
Parcel_se_3	2.27	0.60	−0.68	0.08
Parcel_se_4	2.01	0.54	−0.41	−0.03
Sex (0 = male, 1 = female)	0.53	0.50	−0.13	−1.99
Father–child group (N = 442)				
Positive parenting (pp; scored 1–5)				
Parcel_pp_1	4.14	0.59	−0.71	0.40
Parcel_pp_2	3.81	0.62	−0.60	0.30
Parcel_pp_3	4.07	0.67	−0.83	0.52
Parcel_pp_4	3.97	0.70	−0.78	0.89
Negative parenting (np; scored 1–5)				
Parcel_np_1	2.23	1.04	0.81	0.21
Parcel_np_2	2.00	1.11	1.08	0.48
Parcel_np_3	1.94	1.08	1.17	0.77
Parcel_np_4	2.19	1.06	0.69	−0.18
Parental narcissism (pn; scored 0–16)				
Parcel_pn_1	0.69	0.82	1.12	0.99
Parcel_pn_2	1.33	1.00	0.53	−0.13
Parcel_pn_3	0.76	0.91	1.09	0.76
Parcel_pn_4	0.82	0.94	0.91	−0.04
Parental overvaluation (po; scored 0–3)				
Parcel_po_1	0.80	0.62	0.58	0.06
Parcel_po_2	0.79	0.70	0.92	0.42
Parcel_po_3	1.27	0.88	0.30	−0.91
Parcel_po_4	0.60	0.90	1.46	1.14
Children’s narcissism (cn; scored 0–3)				
Parcel_cn_1	0.66	0.55	1.04	1.44
Parcel_cn_2	0.82	0.71	0.76	0.09
Parcel_cn_3	1.22	0.76	0.37	−0.55
Parcel_cn_4	1.01	0.52	0.56	0.52
Self-esteem (se; scored 0–3)				
Parcel_se_1	1.85	0.69	−0.34	−0.22
Parcel_se_2	2.05	0.59	−0.36	−0.06
Parcel_se_3	2.29	0.59	−0.67	0.11
Parcel_se_4	2.02	0.53	−0.45	0.09
Sex (0 = male, 1 = female)	0.53	0.50	−0.14	−1.99

**Table 2 ijerph-17-05515-t002:** Bivariate correlations among the latent and control variables of the study after estimating the structural equation models, specifying all covariances between them for both the mother–child and father–child groups.

	1.	2.	3.	4.	5.	6.	7.
1. Positive parenting		−0.04	0.03	0.05	−0.04	0.06	−0.07
2. Negative parenting	−0.10		−0.02	0.14	0.12	−0.01	−0.08
3. Parental narcissism	−0.19 **	0.11		0.38	0.16 *	0.03	−0.01
4. Parental overvaluation	0.14 *	0.11	0.46 ***		0.16 *	0.10	−0.03
5. Children’s narcissism	−0.01	0.08	0.20 **	0.08		0.27 ***	−0.19 **
6. Self-esteem	0.15 *	−0.04	0.04	0.07	0.29 ***		−0.02
7. Sex (0 = male, 1 = female)	−0.02	−0.15 *	0.01	−0.04	−0.18 *	−0.04	

Note: Lower diagonal—correlation matrix for the mother–child data (*N* = 494). Upper diagonal—correlation matrix for the father–child data. * *p* < 0.05, ** *p* < 0.01, *** *p* < 0.001.

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
