# Peer review of "The Apple of Daddy’s Eye: Parental Overvaluation Links the Narcissistic Traits of Father and Child"

_ijerph, 2020, doi:10.3390/ijerph17155515_

Round 1
Reviewer 1 Report
The authors investigated the relationship between parental overvaluation and children’s narcissistic trait and found interesting links between maternal narcissism and parenting behaviors and childhood narcissism, and an interesting mediating rote for overvaluation in the relationship between paternal and child narcissism.
I really enjoyed reading this paper; it is particularly pertinent in our times to understand the root causes of narcissism given its distressing among children and college-aged students. The background review was thorough and did a good job of setting up the theoretical motivation for the study.
There are several things I think the authors need to pay more careful attention to:
- General grammatical and sentence construction throughout certain parts of the paper. I found less issues with this in the Methods and Results, but there were numerous sentence construction and grammatical issues in the Introduction and Discussion, e.g., pg. 2, line 85, “Such gaps do not contribute to understand…” is grammatically incorrect. Similarly, pg. 2, line 94, “Consistently with the Social Learning Theory…” also is grammatically incorrect. Pg. 3, lines 96-97 – very poor sentence construction. These problems occur throughout the paper and must be addressed.
- 7 – the sub-section titled Data Analysis is one very long paragraph that needs to be broken up into smaller paragraphs for easier readability and comprehension. This entire section must be reorganized into paragraphs according to the hypotheses, specifying the analysis strategy for each hypothesis.
I enjoyed the discussion surrounding gender differences between mothers’ and fathers’ intergenerational transfer of narcissism to their children. I would also like to see some discussion on the gender differences in narcissism found between male and female children in both parental models.
Author Response
Comments of Reviewer 1
The authors investigated the relationship between parental overvaluation and children’s narcissistic trait and found interesting links between maternal narcissism and parenting behaviors and childhood narcissism, and an interesting mediating role for overvaluation in the relationship between paternal and child narcissism.I really enjoyed reading this paper; it is particularly pertinent in our times to understand the root causes of narcissism given its distressing among children and college-aged students. The background review was thorough and did a good job of setting up the theoretical motivation for the study.
There are several things I think the authors need to pay more careful attention to:
(Query 1)General grammatical and sentence construction throughout certain parts of the paper. I found less issues with this in the Methods and Results, but there were numerous sentence construction and grammatical issues in the Introduction and Discussion, e.g., pg. 2, line 85, “Such gaps do not contribute to understand…” is grammatically incorrect. Similarly, pg. 2, line 94, “Consistently with the Social Learning Theory…” also is grammatically incorrect. Pg. 3, lines 96-97 – very poor sentence construction. These problems occur throughout the paper and must be addressed.
(Response 1). We have reviewed all the above senteces. Moreover, an English speaking proof reader reviewed extensively the manuscript and dealt with all of the all language issues.
(Query 2)The sub-section titled Data Analysis is one very long paragraph that needs to be broken up into smaller paragraphs for easier readability and comprehension. This entire section must be reorganized into paragraphs according to the hypotheses, specifying the analysis strategy for each hypothesis.
(Response 2). Thanks for your note. We followed your suggestions and reorganized the Data Analysis section. Now, each paragraph is related to one objective or hypothesis. Also, we slightly changed the final paragraph of the Introduction section to better introduce the issue of mediation (see p. 3, lines 129-131). However, please, note that to respond adequately to the objectives and hypotheses of a paper in terms of specific analyses, we need to do preliminary analyses that investigate the nature of the data, such as for example the characteristics of normality or non-normality. This, in fact, informs us about the appropriate use of statistics and, in the case of SEM, implies the use of estimators that take into account the characteristics of the data. Furthermore, it is common to report bivariate correlations to ensure the reproducibility of the analyses. Also in this paper, therefore, points 1 and 2 of the analytical steps respond to these preliminary needs rather than to respond directly to the questions posed in the introduction.
(Query 3)I enjoyed the discussion surrounding gender differences between mothers’ and fathers’ intergenerational transfer of narcissism to their children. I would also like to see some discussion on the gender differences in narcissism found between male and female children in both parental models.
(Response 3) Thank you, we are glad you appreciated the discussion surrounding gender differences between mothers’ and fathers’ intergenerational transfer of narcissism to their children. We also briefly discussed the gender differences found between boys and girls (see p. 14).
Reviewer 2 Report
The present study makes an interesting and valuable contribution by disentangling the relationships between parent narcissism, parent style and child narcissism. A cross-sectional design was used with self-report data from a non-clinical sample, which despite limitations appropriately acknowledged by the authors, allowed for a large sample to be obtained and results meaningfully analysed using SEM.
This work contributes to understanding of the development of narcissistic traits and has implications for clinical and other interventions that are likely to target parenting style rather than having a focus on intergenerational transmission of narcissism. It provides a solid foundation for further longitudinal and clinical studies.
My main recommendation for revision is to change the introduction so it more clearly builds evidence for the hypotheses. I think it would be difficult for someone unfamiliar with this field to understand how/why the hypotheses were derived and that could lead to some doubt about the study overall. Cross-sectional studies using SEM to establish pathways need to be driven by strong hypotheses and I'm not sure that comes through. Starting with Ovid's account may work well in a book chapter that has a long section on the history of the study of narcissism, but it didn't work so well here. It was also unclear which ideas were carried over from theoretical work from the 70s/80s/90s and what has come from more recent empirical studies. It would be helpful if this could be made clear. It would also be helpful to make clear what age groups were included in previous research/theory as it is unclear why 4th-5th grade children were selected for the present study and how the present study fits with previous developmental work.
Line 12, change 'build' to 'builds'
Please include the age of the participants in the abstract. It might also help to give an indication of age or grade in the title (e.g. 4th and 5th grade). It needs to be clear from the beginning that this is about pre-adolescent children. It's not until the participants section that age is mentioned. Having details clear in the title and abstract help readers decide whether to continue with the article and assist with article screening for systematic reviews and meta-analyses.
Try to avoid gendered occupation names (workmen, craftsmen).
Abstract, line 25 - should 'cultural' = 'gender'?
Author Response
Comments of Reviewer 2
The present study makes an interesting and valuable contribution by disentangling the relationships between parent narcissism, parent style and child narcissism. A cross-sectional design was used with self-report data from a non-clinical sample, which despite limitations appropriately acknowledged by the authors, allowed for a large sample to be obtained and results meaningfully analysed using SEM.This work contributes to understanding of the development of narcissistic traits and has implications for clinical and other interventions that are likely to target parenting style rather than having a focus on intergenerational transmission of narcissism. It provides a solid foundation for further longitudinal and clinical studies.
(Query 1)My main recommendation for revision is to change the introduction so it more clearly builds evidence for the hypotheses. I think it would be difficult for someone unfamiliar with this field to understand how/why the hypotheses were derived and that could lead to some doubt about the study overall. Cross-sectional studies using SEM to establish pathways need to be driven by strong hypotheses and I’m not sure that comes through. Starting with Ovid's account may work well in a book chapter that has a long section on the history of the study of narcissism, but it didn’t work so well here. It was also unclear which ideas were carried over from theoretical work from the 70s/80s/90s and what has come from more recent empirical studies. It would be helpful if this could be made clear. It would also be helpful to make clear what age groups were included in previous research/theory as it is unclear why 4th-5th grade children were selected for the present study and how the present study fits with previous developmental work.
(Response 1)Thank you for this usefull suggestion. We edited the Introduction to address your concerns. We remove the Ovid’s account. We added more information (i.e., age) about the sample used in previous studies, to underline the importance of inestigating narcissism at an earlier age, such as late childhood.
(Query 2)Line 12, change 'build' to 'builds'
(Response 2)The abstract has been modified and this typo edited.
(Query 3)Please include the age of the participants in the abstract. It might also help to give an indication of age or grade in the title (e.g. 4th and 5th grade). It needs to be clear from the beginning that this is about pre-adolescent children. It's not until the participants section that age is mentioned. Having details clear in the title and abstract help readers decide whether to continue with the article and assist with article screening for systematic reviews and meta-analyses.
(Response 3)We agree with your concern and we added the participants’ age to the abstract. Age information was not added to the tile not to make it too long.
(Query 4)Try to avoid gendered occupation names (workmen, craftsmen).
(Response 4). Thank you for your suggestion. We have replaced the two terms with “laborer and artisan”.
(Query 5)Abstract, line 25 - should 'cultural' = 'gender'? The description of participants requires information on how the sample was obtained e.g., rural, urban, large scale study, assortment of other studies, etc. This is essential information to focus the reader on the relevant sample.
(Response 5)The abstract was completely revised. We added this information.
Other revisions by the authors
We have added an author, Doct. Annarita Milone, as she has provided consistent support in the revision of the first version of the manuscript.
As requested, we confirm the author list and the corresponding affiliations. Authors' names and email addresses in the manuscript should are consistent with that in the system.